# Exploring Plant Growth-Promoting Traits of Endophytic Fungi Isolated from *Ligusticum chuanxiong* Hort and Their Interaction in Plant Growth and Development

**DOI:** 10.3390/jof10100713

**Published:** 2024-10-12

**Authors:** Qing Wang, Xinyu Zhang, Qiqi Xie, Jiwen Tao, Yujie Jia, Yirong Xiao, Zizhong Tang, Qingfeng Li, Ming Yuan, Tongliang Bu

**Affiliations:** 1College of Life Sciences, Sichuan Agricultural University, Ya’an 625014, China; wangqing2024822@163.com (Q.W.); zxy9379@163.com (X.Z.); xieqq0972@163.com (Q.X.); taojiwen2000@163.com (J.T.); yuansheng0262@gmail.com (Y.J.); qingfeng.li@sicau.edu.cn (Q.L.); yuanming@sicau.edu.cn (M.Y.); 13941@sicau.edu.cn (T.B.); 2Sichuan Agricultural University Hospital, Ya’an 625014, China; xxyyrr01@163.com

**Keywords:** fungal endophytes, plant growth promotion, germination indices, growth indicators, physiological characteristics

## Abstract

Endophytic fungi inhabit various plant tissues and organs without inducing evident disease symptoms. They can contribute positively to the growth of plants, bolster plants resilience to environmental and biological stresses, and facilitate the accumulation of secondary metabolites. These microbial resources possess significant developmental and utilization value in various applications. Hence, this study focused on exploring the plant growth-promoting (PGP) traits of 14 endophytic fungi from *Ligusticum chuanxiong* Hort (CX) and elucidating the effects and mechanisms that facilitate plant growth. According to PGP activity evaluation, the majority of strains demonstrated the capacity to produce IAA (78.57%), siderophores (50.00%), ammonia (35.71%), potassium solubilization (21.43%), nitrogen fixation (57.14%), and phosphate solubilization (42.86%). Further investigations indicated that the levels of IAA ranged from 13.05 to 301.43 μg/mL, whereas the soluble phosphorus levels ranged from 47.32 to 125.95 μg/mL. In cocultivation assays, it was indicated that *Fusarium* sp. YMY5, *Colletotrichum* sp. YMY6, *Alternaria* sp. ZZ10 and *Fusarium* sp. ZZ13 had a certain promoting effect on lateral root number and fresh weight of tobacco. Furthermore, ZZ10 and ZZ13 significantly enhanced the germination potential, germination index, and vigor index of tobacco seeds. The subsequent potted trials demonstrated that the four endophytic fungi exhibited an enhancement to growth parameters of tobacco to a certain extent. ZZ10 and ZZ13 treatment had the best promotion effect. Inoculation with YMY5 increased the chlorophyll a and total chlorophyll content. ZZ10 and ZZ13 treatment remarkably increased the net photosynthetic rate, soluble sugars and soluble protein content, catalase and peroxidase activities, and lowered malondialdehyde content in tobacco leaves. In addition, YMY5 remarkably elevated superoxide dismutase activities. ZZ13 upregulated the expression of growth-related gene. Among them, ZZ13 had a better growth-promoting effect. In conclusion, these endophytic fungi possessing multi-trait characteristics and the capacity to enhance plant growth exhibit promising potential as biofertilizers or plant growth regulators.

## 1. Introduction

Endophytic fungi reside in the intercellular and extracellular regions of healthy plants and usually form a mutualistic relationship with their hosts [1]. The host plants provide protection, nutrition, and shelter for endophytic fungi, while endophytic fungi not only promote plant growth but also boost the plant tolerance against ecological stress by synthesizing diverse secondary metabolites [2]. Abundant endophytes can be isolated from the vegetative organs of various plant types, such as woody and herbaceous plants [3,4]. Almost every plant contains one or more endophytes. In recent years, the crucial role of endophytic fungi in ecosystem dynamics has been increasingly acknowledged, and they have great research and development potential in agriculture and industrial applications.

Through a combination of direct and indirect pathways, endophytic fungi facilitate plant growth. The direct mechanism involves the stimulation or induction of plants to synthesize specific bioactive substances, like the production of phytohormones, siderophore, ACC (1-aminocyclopropane-1-carboxylic acid) deaminase, nitrogen fixation, phosphorus solubilization, and potassium solubilization [5]. The indirect mechanism involves the secretion of specific substances to suppress pathogens in the plant rhizosphere or enhance the plant’s innate defense, such as producing antibiotics, hydrogen cyanide, competition, and antagonism [6]. Endophytic fungi often produce phytohormones like gibberellin (GA), indole-3-acetic acid (IAA), salicylic acid (SA), and cytokinin (Cks) to promote plant growth [7,8]. One of the most common phytohormones is IAA, with low concentrations facilitating primary root elongation and high levels promoting lateral root number, reducing primary root length and increasing root hair growth [9]. In addition, endophytic fungi can facilitate the acquisition capabilities of inorganic elements by plants, including nitrogen, phosphorus, potassium, zinc, and iron, thereby improving soil nutrient accessibility and reducing the requirement for biofertilizers [10]. Endophytic fungi obtained from Sunchoke (*Helianthus tuberosus* L.), Siam weed (*Chromolaena odorata*), ginger (*Zingiber officinale*), and stemona (*Stemona sessilifolia* (Miq.) Miq.) exhibited the ability to synthesize IAA, ammonia, hydrogen cyanide, siderophore, and extracellular enzymes [11]. Endophytic fungi can increase the accumulation of plant secondary metabolite, which may be synthesized by either host plant or endophytic fungi [12]. Certain endophytic fungi found in medicinal plant can even boost the accumulation of medicinal active ingredients. Chen et al. [13] indicated that endophytic fungi isolated from *Bletilla striata* (Thunb.) can enhance the activities of pivotal enzymes in the phenylpropane metabolic pathway, thereby facilitating total phenol synthesis. Additionally, fungal endophytes can boost host plant’s tolerance to challenging environmental conditions like drought, high salinity, cold temperatures, heat temperatures, and heavy metal stress [14,15,16]. Endophytic fungi from buckwheat (*Fagopyrum Tataricum*) can increase the metabolite content and promote growth of their host plants [17]. Similarly, endophytic fungal strains from the *Anoectochilus* and *Ludisia orchids* facilitated plant growth and enhanced the yield of bioactive metabolites [18]. Endophytic fungi can also enhance plant seed germination [19], regulate root development [20], facilitate nutrient uptake [21], modulate phytohormone interactions, and regulate photosynthesis [22]. In the context of sustainable agriculture, the development of beneficial endophytic fungi is crucial.

*Ligusticum chuanxiong* Hort (CX), extensively used in traditional Chinese medicine, is primarily cultivated in Sichuan Province, China. It has pharmacological effects on enhancing blood circulation, relieving rheumatism and pain, and treating cerebrovascular and cardiovascular diseases [23,24,25]. A previous study demonstrated that CX hosted a wide variety of endophytic fungi, and these fungi had antioxidant and antibacterial activities [26]. *Penicillium oxalicum*, an endophytic fungus from CX, exhibited properties that protected DNA and enhanced stress tolerance in *Caenorhabditis elegans* [27]. Additionally, endophytic fungi also contribute significantly to influencing plant growth and development. Kang et al. [28] indicated that three fungi from CX not only supported the growth of shoots, stem nodes, and internodes of stem nodes of CX but also boosted the activity of plant peroxidase, catalase, and phenylalanine ammonia lyase. At present, the effect of endophytic fungi from CX on plants is still limited, and its functional significance is still unclear. The present study aimed to analyze plant growth-promoting characteristics in vitro of endophytic fungi from CX, including IAA production, nitrogen fixation, phosphate solubilization, potassium solubilization, ammonia production, and siderophore production. Selected fungal endophytes as microbial inoculants to enhance growth performance in tobacco were subsequently assessed. Inoculating tobacco with endophytic fungi to determine whether this inoculation can enhance seed vitality, improve plant growth indicators, promote root development, and alter its physiological functions provides an important reference for screening excellent growth-promoting strains.

## 2. Materials and Methods

### 2.1. Endophytic Fungi and Plant Materials

The endophytic fungi isolated from CX by the early research group were utilized as the experimental strains [26]. The 14 isolates were cultured and maintained on Potato Dextrose Agar (PDA) medium (containing potato 200 g/L, glucose 20 g/L, agar 20 g/L), including *Rigdoporus vinctus* ZZ2, *Rigdoporus vinctus* ZZ3, *Simplicillium* sp. ZZ5, *Cladosporium cladosporioides* ZZ9, *Alternaria alternata* ZZ10, *Alternaria alternata* ZZ11, *Fusarium fujikuroi* ZZ12, *Fusarium tricinctum* ZZ13, *Penicillium oxalicum* YMG1, *Fusarium asiaticum* YMY5, *Colletotrichum camelliae* YMY6, *Sordariomycetes* sp. YMJ10, *Fusarium proliferatum* YMJ11, and *Colletotrichum fructicola* YMJ13. The common variety (NC89) of tobacco (*Nicotiana tabacum* L.), supplied by the Department of Botany at the College of Life Sciences, Sichuan Agricultural University, was selected as the model plant.

### 2.2. Evaluation of PGP Traits of Endophytic Fungi

#### 2.2.1. Analysis of Indole-3-Acetic Acid (IAA)

The synthesis of IAA was assessed based on Salkowski’s method as outlined by Khalil et al. [29] with slight adjustments. The 14 strains of fungi selected were inoculated into Potato Dextrose Broth (PDB) medium (containing potato 200 g/L, glucose 20 g/L) with an additional of 200 mg/mL of tryptophan, followed by incubation at 28 °C for 7 d. Controls consisted of PDB medium that was not inoculated. After incubation, each culture was centrifuged at 6000 rpm for 30 min, and 1 mL of culture supernatant was blended with same volume of Salkowski’s reagent (containing 60 mL of concentrated sulfuric acid, 100 mL of ddH_2_O, and 3 mL of 0.5 mol/L FeCl_3_). If a pink color appeared, it signified that IAA was present in the fermentation broth. The OD was recorded at 530 nm with a Microplate Reader (Multiskan Sky, Thermo Fisher Scientific, Shanghai, China). The standard curve was made by 1000 μg/mL IAA standard solution as the mother liquor. Each treatment was replicated three times.

#### 2.2.2. Phosphate Solubilization

Phosphate-solubilizing ability of 14 endophytic fungi was assessed using National Botanical Research Institute’s phosphate growth (NBRIP) medium [30]. NBRIP agar medium (glucose 10 g/L, Ca_3_(PO_4_) 25 g/L, MgCl_2_·6H_2_O 5 g/L, MgSO_4_·7H_2_O 0.25 g/L, KCl 0.2 g/L, (NH_4_)_2_SO_4_ 0.1 g/L, agar 15 g/L, pH 7.0) was prepared. Endophytic fungi were grown on NBRIP agar medium for 7 d at 28 °C to observe the formation of a transparent zones surrounding colonies. The concentration of soluble phosphorus was subsequently determined. Fungal strains were inoculated into NBRIP liquid medium. Afterward, the supernatants were collected by centrifugation at 10,000 rpm for 10 min, followed by analysis using the molybdenum antimony colorimetric method [31]. The concentration of phosphates was quantified by using potassium dihydrogen phosphate as the reference standard curve. Three replications were made.

#### 2.2.3. Nitrogen Fixation

The nitrogen fixation capability of the strains was assessed using nitrogen-free culture medium (5.0 g/L glucose, 0.2 g/L KH_2_PO_4_, 0.2 g/L MgSO_4_·7H_2_O, 0.2 g/L NaCl, 0.1 g/L CaSO_4_·2H_2_O, 5 g/L CaCO_3_, 15 g/L agar, pH = 7.0–7.2). The 14 fungal strains were inoculated on nitrogen-free medium at 28 °C for 7 d [32]. The growth of fungal colonies served as the standard for evaluating their ability to fix nitrogen. Each strain was repeated three times.

#### 2.2.4. Siderophores Production

The siderophores production of endophytic fungi was detected based on chrome azurol sulfonate (CAS) medium as described by [33]. The CAS medium contained (per 100 mL) 20% sucrose 1 mL, 10% casamino acid 3 mL, 1 mM CaCl_2_ 100 µL, CAS dye solution 5 mL (containing chrome azurol S 3 μg, hexadecyltrimethylammonium bromide 3.75 μg, 1 mM FeCl_3_ 0.5 mL, and distilled water 4.5 mL), phosphate buffer (pH 6.8–7.0) 5 mL, and agar 2 g. The phosphate buffer and CAS dye solution were sterilized individually, and then were mixed into the CAS medium. After inoculating 14 endophytic fungi into CAS medium for a duration of 5–7 d at 28 °C, the color transition from blue to orange or purple-red was observed, indicating the production of the siderophore [34]. Each group was performed three times.

#### 2.2.5. Potassium Solubilization

The Aleksandrov agar medium (containing 10 g/L glucose, 0.2 g/L Na_2_HPO_4_, 0.2 g/L MgSO_4_·7H_2_O, 0.2 g/L NaCl, 0.2 g/L CaSO_4_·2H_2_O, 5 g/L CaCO_3_, 15 g/L agar, 2.5 g/L potassium feldspar powder, pH 7.2) was utilized to evaluate the potassium-dissolving capability of endophytic fungi [35]. The 14 endophytic fungi were incubated on the Aleksandrov agar medium at 28 °C for 7 d. The observation of a transparent halo surrounding the fungal colony was thought to have the potassium-dissolving activity. Each group was performed three times.

#### 2.2.6. Ammonia Production

The ammonia production activity was evaluated following the methodology proposed by Khalil et al. [29] with slight adjustments. The 14 endophytic fungi were cultivated in PDB medium at 28 °C in a shaker (180 rpm) for 7 d. The culture medium was centrifuged at 12,000 rpm for 10 min, and 1.7 mL supernatant was mixed with 0.2 mL Nessler’s reagent (K_2_HgI_4_ and NaOH). The transformation of color from yellow to brown denoted the presence of ammonia, with a darker shade suggesting a higher ammonia concentration. PDB medium without fungal inoculations was used as the control group. Each group was performed three times.

### 2.3. Cocultivation of Endophytic Fungal Strains with Tobacco

The preliminary assessment of how endophytic fungi affect tobacco plant development was conducted through cocultivation experiments on Murashige and Skoog (MS) medium. The culture medium consisted of 2.68 g/L MS medium (Haibo Biotechnology, Qingdao, China) supplemented with 30 g/L of sucrose and 7.5 g/L agar, and pH = 5.8–6.2. Before the cocultivation process, tobacco seeds first were by immersion in 75% ethanol for 30 s, rinsed three times using sterile water, then soaked 10% sodium hypochlorite for 4 min, and finally rinsed five times with sterile water. Afterward, sterilized seeds were planted on MS agar medium and placed in a cultivation chamber at 24 °C under a 16 h light/8 h dark cycle. At the end of 7 d, 6–8 seedlings were transferred to a new MS medium, and endophytic fungal colonies (5.5 mm in diameter) were inoculated on agar plates 6 cm below the tobacco roots tips for 12 d and 17 d. Non-inoculated group were used as controls [36]. The plates were positioned vertically to promote root growth along the agar surface. After a duration of 12 d, these parameters were evaluated: fresh weight, length, and the quantity of lateral roots.

### 2.4. In Vitro Germination and Seedling Vigor Test

Seeds were sterilized as described above. Spore suspensions are prepared as follows: first, endophytic mycelium was added on the PDA medium and incubated at 28 °C for 7 days; second, fungal spore suspensions were prepared by adding 10 mL of ddH_2_O, followed by scraping spores from the surface; and third, the spore suspension was mixed, centrifuged at 5000 rpm for 20 min, the supernatant was discarded, and suspensions were rinsed with sterile water (repeat this step three times). Later, a spore suspension of 1 × 10^7^ spores/mL was obtained by adding ddH_2_O water. Twenty tobacco seeds were soaked with 1 mL (1 × 10^7^ spores/mL) of spore suspension for 24 h, and then they were placed in a petri dish containing filter paper. The control samples were sterilized seeds that had been soaked in sterile water for an equivalent period. The seeds were kept moist by spraying 5–6 mL of sterilized distilled water on filter paper. The quantity of sprouted seeds and the time taken for germination of each seed were recorded. After 13 d, we assessed both the root length and fresh weight of seedlings. The germination characteristics were evaluated as described by Wang and Chen [37]:Germination rate (GR) = (number of germinated seeds at 13 days/total number of seeds) × 100%; 
Germination potential (GP) = (number of germinated seeds at 8 days/total number of seeds) × 100%; 
Germination index (GI) = ∑(Gt/Dt) × 100%; 

Gt: number of germinated seeds at specific days and Dt: number of specific days
Vigor index (VI) = germination rate (%) × seedling length

### 2.5. Pot Experiment

The capacity of endophytic fungi to enhance plant growth was evaluated in pot conditions. Fungi were cultivated in PDB medium and incubated for 7 d at 28 °C with a shaking condition of 180 rpm. Then, the fermentation broth was filtered to harvest the mycelium, washed with sterile ddH_2_O, and 30 g of fungi mycelium was mixed with 1000 g sterile soil (high-temperature steam sterilization at 121 °C, 0.1 MPa for 20 min). Tobacco seeds were sown in individual pots (13.5 cm height, top dimensions of 16 cm × 16 cm and bottom dimensions of 11 cm × 11 cm) and grown at 24 °C (14 h light/10 h dark). The control group treatments (CK) involved only sterile soil without the presence of endophytic fungi. The endophytic fungi group and the control group had 10 pots. After 50 d, the growth of tobacco plants was observed, and the plant height, fresh weight, and maximum leaf area were measured. Additionally, colonization and interrelated physiological and biochemical index were detected.

### 2.6. Determination of Fungus Colonization in Tobacco Roots

The colonization of endophytic fungi was observed utilizing the approach outlined by Gateta et al. [38] with certain modifications. Briefly, tobacco roots were cleaned with tap water, sliced into 1 cm fragments, added to a 10% KOH solution, and exposed to a 90 °C water bath for 30 min. Subsequently, roots were treated in 10% H_2_O_2_ at room temperature for 30 min, acidified in 5% glacial acetic acid for 30 min, and stained with an acetic glycerin solution (consisted of 0.02 g of trypan blue, 50 mL of glycerin, 5 mL of 1% hydrochloric acid, and 45 mL of distilled water). Then, roots were faded in glycerin acetate solution (prepared by mixing equal volumes of glycerol, acetic acid, and ddH_2_O) for 5 h. The root samples were analyzed for endophytic fungi colonization under a microscope (BX53, Olympus Corporation, Tokyo, Japan). Three root segments (3 cm length each) per plant were screened.

### 2.7. Analysis of Chlorophyll Content and Photosynthesis Parameters

The photosynthetic pigments were obtained by grinding 0.02 g of fresh leaf in 2 mL of 95% ethanol, followed by adding another 10 mL of 95% ethanol. The sample was kept at 4 °C for 2 h under dark conditions and then centrifuged at 4000 rpm. The spectrophotometer was utilized to assess the OD at both 663 nm and 645 nm. The chlorophyll content was evaluated as described by Arnon [39]. Each group was performed three times.

The fluorescence imaging system GFS-3000 (WALZ, Nuremberg, Germany) was utilized to evaluate the transpiration rate (Tr), stomatal conductance (Gs), intercellular CO_2_ concentration (Ci), and net photosynthetic rate (Pn) in tobacco leaves. Three plants in each treatment were tested, and a leaf was chosen from an identical location on each plant. Each group was performed three times.

### 2.8. Analysis of Soluble Protein and Soluble Sugar Content

Brilliant Blue G-250 method was used to estimate soluble protein [40]. First, 0.1 g of fresh tobacco leaves were added to 0.9 mL 0.9% normal saline, and a 10% tissue homogenate was prepared in ice bath conditions. Centrifugation of the homogenates was carried out at 2500 rpm for 10 min, followed by measuring the absorbance of the supernatant at 595 nm. Each group was performed three times.

Analysis of soluble sugar content was carried out according to the anthrone colorimetry method. The homogenate was prepared by grinding 0.1 g of tobacco leaves in 1 mL of ddH_2_O. Afterward, the mixture was put into a boiling water bath for 10 min, then cooled and centrifuged at 4000 rpm for 10 min. The supernatant was diluted with distilled water to 10 times. These diluted solutions were analyzed by the kit from Nanjing Jiancheng Biological Engineering Institute (Nanjing, China). Each group was performed three times.

### 2.9. Quantification of Antioxidant Enzyme Activity and Oxidative Stress Markers

First, 0.1 g of tobacco leaves were ground with 0.9 milliliters of 0.2 M phosphoric acid buffer in ice bath conditions. Following centrifugation at 3500 rpm for 10 min, the supernatant was reserved for subsequent analysis. The activity of peroxidase (POD), catalase (CAT), and superoxide dismutase (SOD) and the content of malondialdehyde (MDA) were carried out utilizing the specific reagent kit from Nanjing Jiancheng Biological Engineering Institute (Nanjing, China). Each group was performed three times.

### 2.10. Detection of Expression Levels of Growth-Related Genes by Real-Time PCR (RT-qPCR)

To explore the molecular mechanisms by which endophytic fungi regulate tobacco growth, the expression levels of plant hormones and growth-related genes in leaves were examined by RT-qPCR. The expression levels of cell division-related genes, auxin (IAA), gibberellin (GA_3_), salicylic acid (SA), and brassinosteroids (BR) biosynthesis-related genes were analyzed. Extraction of total RNA from 0.1 g of frozen plant samples was carried out using an RNA extraction kit (Vazyme, Nanjing, China), and the NanoDrop 2000C system (ThermoFisher, Shanghai, China) was used to detect the concentration and purity of the RNA samples (ThermoFisher). The PrimeScript™ RT reagent Kit was used to synthesize the first-strand cDNA (Tiangen, Beijing, China). Afterward, RT-qPCR analysis utilized the SuperReal PreMix Plus (SYBR Green) (Tiangen, Beijing, China). Each group was performed three times. The primer sequences for the quantitative analysis were synthesized by Tsingke Biotechnology Co., Ltd. (Beijing, China). The primers sequences of the relevant genes are outlined in Table 1. The relative expression of each gene was assessed via the 2^−∆∆Ct^ method, taking *β-Actin* as the reference gene.

### 2.11. Statistical Analyses

The data were processed and analyzed through IBM SPSS 24, while graphical representations were created using Origin 2021. Before analysis, the data were tested for normality and homogeneity of variance. Significant differences among treatments were evaluated by analysis of variance (ANOVA), and Tukey’s HSD post-hoc test was used for multiple comparisons at α = 0.05. In the bar chart, different lowercase letters represent statistically significant variations among treatment groups (*p* < 0.05), while the same letters denote that there were not statistically significant between each group (*p* > 0.05). The results are presented in terms of mean ± standard deviation (SD). 

## 3. Results

### 3.1. In Vitro Assessment for PGP Traits of Endophytic Fungi

#### 3.1.1. Production of Indole-3-Acidic Acid (IAA)

The synthesis of plant hormones by endophytic fungi is pivotal to fostering plant growth. In this study, the production of IAA, the one of most common plant hormones synthesized by endophytic fungi, was determined. Overall, 78.57% of endophytic fungi displayed varying degrees of pink coloration, whereas ZZ5, ZZ9, and YMG1 exhibited no notable color differences relative to the control group (Figure 1A). The qualitative analysis results revealed the levels of IAA production from 13.05 to 301.43 μg/mL (Figure 1B). YMJ13 exhibited the highest IAA production at 301.43 μg/mL, followed by YMY5 at 96.02 μg/mL. 

#### 3.1.2. Qualitative and Quantitative Analysis of Phosphate Solubilization

The 14 endophytic fungal strains were inoculated on inorganic phosphate NBRIP medium to assess their phosphate-dissolution potential based on the formation of halo zone surrounding the colonies. The analysis demonstrated that only six isolates, including ZZ9, YMG1, YMY5, YMY6, ZZ10, and ZZ13, exhibited the formation of clear zones, indicating their ability to solubilize phosphate (Figure 2A). When the NBRIP–broth medium was used to assess the phosphorus solubilization capacity of the six strains, the higher amounts of phosphate solubilization was observed with strains YMG1, ZZ9, and ZZ13, which were 125.95, 111.95, and 107.69 μg/mL, respectively (Figure 2B). 

#### 3.1.3. Siderophores and Nitrogen Fixation

The production of siderophores was confirmed by a color shift in the CAS medium from blue to orange or purple red. The results suggested that 50.00% of endophytic fungal strains exhibited a halo zone surrounding mycelia plugs, demonstrating their potential to produce siderophores (Table 2). Among them, strains YMJ13, ZZ10, YMJ11, ZZ13, and ZZ11 exhibited clear orange or yellow halo zones on the CAS medium. Strains ZZ9 and YMG1 produced distinct dark purplish red colonies on the CAS medium (Figure 3A). 

Another crucial PGP trait that can influence plant growth is nitrogen fixation. The study revealed that 57.14% of endophytic fungi, including ZZ2, ZZ9, ZZ10, ZZ11, ZZ13, YMG1, YMY5, and YMY6, could successfully grow on nitrogen-free culture medium (Figure 3B). This finding indicates that these specific strains possessed nitrogen-fixing activity. 

#### 3.1.4. Potassium Solubilization and Ammonia Production

In this study, only three endophytic fungal strains, namely ZZ2, ZZ9, and YMG1, grew normally on the medium plate supplemented with potassium feldspar powder and produced obvious halo zones, indicating their ability to solubilize potassium (Table 2, Figure 3C). 

The production of ammonia can impact the recycling of nitrogen sources and fertility levels in soil. Additionally, the acidic substances during ammonia production can aid in neutralizing alkaline substances in soil and regulating soil pH levels. Following the addition of Nessler’s reagent, the broth media of five fungal strains (YMJ13, ZZ10, ZZ13, YMY5, and YMY6) exhibited color changes ranging from pale yellow to brownish color (Table 2, Figure 3D). Strains ZZ10, YMY5, and YMY6 exhibited the darker coloration, suggesting that these strains possessed the higher capacity of ammonia production.

### 3.2. Cocultivation of Endophytic Fungal and Tobacco Seedlings

Among the fourteen endophytic fungi, four successfully grew on the MS agar medium, and phenotypic improvements in plant growth were observed (Figure 4A,B), including YMY5, YMY6, ZZ10, and ZZ13. Therefore, these four endophytic fungal strains were co-cultivated with tobacco seedlings to their potential plant-promoting activity. The root structure and fresh weight of tobacco seedlings were measured after co-culturing with endophytic fungi for 12 d. The growth of primary roots of tobacco seedlings co-cultured with endophytic fungi showed a reduction compared to the control seedlings, with the YMY5 and ZZ10 treatments showing the most significant effects (*p* < 0.05) (Figure 4C). The number of lateral roots in tobacco increased significantly after inoculation treatment (Figure 4E). Furthermore, we observed strains YMY5, YMY6, and ZZ10 significantly increased the fresh weight of tobacco seedlings (*p* < 0.05) (Figure 4D). Seedlings inoculated with endophytic fungi had larger leaves and a greater number of lateral roots, which likely contributed to the higher fresh weight of seedlings compared to the control group.

### 3.3. Effects of Endophytic Fungi on Tobacco Seed Viability

To investigate the impact of strain YMY5, YMY6, ZZ10, and ZZ13 on seed germination, tobacco seeds were soaked with the spore suspension of strains. The results indicated that the treatment with the spore suspension of four strains had no notable change on the germination rate of tobacco seeds relative to the control group (Figure 5A). However, the germination potential, germination index, and vigor index of tobacco seeds in the ZZ13 and ZZ10 treatment groups revealed a significant enhancement (*p* < 0.05) (Figure 5). The germination potential, germination index, and vigor index of the ZZ10-inoculated seedlings increased remarkably by 67.12%, 23.88%, and 75.51%, while those treated with ZZ13 increased by 63.91%, 22.49%, and 92.46%, respectively. In summary, ZZ10 and ZZ13 showed considerable potential in promoting seed vigor and seedling growth. 

### 3.4. The Effect of Endophytic Fungi on Tobacco Growth in Pot Experiments

To investigate the impacts of YMY5, YMY6, ZZ10, and ZZ13 on tobacco growth promotion, we further measured the growth parameters of tobacco in a pot for 50 d, as illustrated in Figure 6. Tobacco treated with endophytic fungi exhibited an enhancement in both overall growth and biomass (Figure 6A,B). Among them, the seedlings inoculated with ZZ10 and ZZ13 showed the most significant enhancements in the length of the whole plant, which increased by 138.57% and 204.57%, respectively (Figure 6C). Furthermore, ZZ10 and ZZ13 treatments markedly elevated the plant weight of tobacco seedlings (*p* < 0.05) (Figure 6D), particularly the ZZ10-treated group. These strains could also promote the maximum leaf area of tobacco relative to the control group (Figure 6E). Tobacco seedlings treated with ZZ10 and ZZ13 exhibited a maximum leaf area of 4321 and 4512.97 cm^2^, and the increasing rate was 237.10% and 252.07% separately compared to the control. These results indicated that YMY5, YMY6, ZZ10, and ZZ13 in pot experiment promoted better growth in tobacco seedlings compared to the non-inoculated control. ZZ10 and ZZ13 treatment had the best growth of tobacco plants.

To determine the capacity of endophytic fungi to colonize tobacco roots, the colonization was checked using a microscope after staining. The surface of roots from potted seedlings treated with endophytic fungi showcased a dense network of fungal hyphae, indicating a robust symbiotic relationship established through plant–microbe interaction. Tobacco seedlings uninoculated with endophytic fungi did not demonstrate any evident growth of fungal hyphal inside their tissues (Figure 6B). 

### 3.5. Effects of Endophytic Fungi on Chlorophyll Content and Photosynthesis Parameters of Tobacco

To analyze the influence of inoculating tobacco with selected endophytic fungi on photosynthesis, we measured several photosynthetic parameters in 50-day-old plants in pots following inoculation. The content of chlorophyll a of the tobacco leaves inoculated with YMY5 was the highest, showing an increase of 71.53% relative to the control group. There was a minor significant difference in the chlorophyll b content among different treatments. The total chlorophyll content of tobacco plants was notably elevated by four endophytic fungi, with the highest increase observed in plants treated with YMY5 at 44.57%. To conclude, YMY5 showed a more pronounced increase in chlorophyll content compared to other strains.

Strains YMY6, ZZ10, and ZZ13 could increase the net photosynthetic rates (Pn) and transpiration rate (Tr) (Figure 7D). Inoculation treatment did not have a significant effect on stomatal conductance (Gs) and transpiration rate (Tr) with respect to control plants. After ZZ13 treatment, the Gs, Pn, and Tr of tobacco leaves reached the maximum. Moreover, the intercellular CO_2_ concentration (Ci) of tobacco leaves decreased after inoculation compared to the control group. However, the difference was not significant compared to the control group.

### 3.6. Effects of Endophytic Fungi on the Content of Soluble Sugar and Soluble Protein of Tobacco

Soluble sugar and soluble protein are important nutrients and physiological active substances in plants, which are critical for plant growth, development and physiological function regulation. Following inoculation with four endophytic fungi for 50 d, the content of soluble sugars and soluble proteins of potted tobacco were measured. The content of soluble sugar of tobacco inoculated with YMY5, ZZ10 and ZZ13 was significantly increased, with increases of 154.55%, 189.38% and 298.87%, respectively (Figure 8A). Similarly, inoculation with these three endophytic fungi elevated the soluble protein content in the leaves of tobacco plants (Figure 8B). The soluble protein content of tobacco inoculated with YMY5, ZZ10 and ZZ13 was significantly increased, with increases of 78.82%, 18.04% and 34.50% compared with the control, respectively. Overall, the inoculation with YMY5, ZZ10 and ZZ13 had the capacity to elevate the levels of soluble sugars and soluble proteins of tobacco.

### 3.7. Analysis of Endophytic Fungi on the Antioxidant Properties of Tobacco

The activity of antioxidant enzymes like SOD, CAT, POD, and MDA was modulated in tobacco leaves inoculated with four endophytic fungi under pot cultures. YMY5 significantly increased SOD activity of tobacco. The other inoculation groups had a minor effect on SOD activity (Figure 9A). The POD activity of tobacco significantly elevated after inoculation (*p* < 0.05) (Figure 9B). Compared with control group, the tobacco leaves treated with YMY5, YMY6, ZZ10, and ZZ13 showed significant increases in POD activity of 71.31%, 74.74%, 78.16%, and 78.63%, respectively. The CAT activity inoculated with four endophytic fungi showed a significant enhancement (*p* < 0.05) (Figure 9C), and ZZ13 increased CAT activity by 220.28%. The MDA content of tobacco treated with ZZ10 and ZZ13 exhibited a marked decrease, with reductions of 43.25% and 51.97%, respectively (Figure 9D). 

### 3.8. Effects of Endophytic Fungi on the Expression of Tobacco Growth-Related Genes

To further understand plant growth-promoting mechanisms of four endophytic fungi at the molecular level, we analyzed the effects of fungal inoculation on the expression of various potential growth-related genes in tobacco leaves under pot cultures. Endogenous hormone-related and cell division-related genes expression were analyzed, including *NtCYCD3*, *NtYUCCA8*, *NtARF6*, *NtARF16*, *NtGA3ox-2*, *NtDWF4*, *NtBIN2,* and *NtICS.* As shown in Figure 10A, the expression of cell cycle-related genes *NtCYCD3* was markedly upregulated after inoculation with YMY5, ZZ10, and ZZ13 (*p* < 0.05). The expression level of *NtCYCD3* in ZZ13-treated samples increased by 137.70%. ZZ13 also significantly modulated the expression of auxin synthesis genes and auxin response factors (*p* < 0.05) (Figure 10B–D). The expression of IAA biosynthetic gene *NtYUCCA8* reached the highest level in the tobacco treated with ZZ10, which was increased by 279.21%. The expression levels of auxin response genes *NtARF6* and *NtARF16* were significantly upregulated by 380.93% and 408.68% in the ZZ13 group, respectively. Also, ZZ13 markedly elevated the expression levels of the brassinosteroid (BR) synthesis and signal transduction-related genes (*p* < 0.05) (Figure 10E,F). In ZZ13, the expression of *NtDWF4* increased by 622.23%, while *NtBIN2* increased by 184.81%. The expression of the gibberellin (GA_3_) biosynthesis gene *NtGA3ox-2* was notably enhanced by 171.36% in the YMY5 treatment group (Figure 10G). YMY5, YMY6, ZZ10, and ZZ13 significantly upregulated the expression of salicylic acid synthesis gene *NtICS* (Figure 10H). ZZ13 markedly enhanced the expression of *NtICS* by 579.85%. The findings indicated that these four endophytic fungi from CX could enhance plant growth and regulate plant defense by upregulating genes associated with cell cycle, IAA, GA_3_, SA, and BR biosynthesis.

## 4. Discussion

Endophytic fungi improve plant growth by secreting plant hormone, promoting mineral absorption, and antagonizing pathogens. These beneficial fungi have gained attention for their potential as biofertilizers. *Ligusticum chuanxiong* Hort, a widely used Chinese medicine, harbors a significant abundance of endophytic fungi. Endophytic fungi have been shown to be very beneficial to plant growth. The study evaluated 14 endophytic isolated from CX for their plant growth-promoting traits in vitro. The results indicated that 78.57% of the endophytes exhibited IAA production, and the concentration of IAA was between 13.05 ± 0.55 and 301.43 ± 4.61 μg/mL (Figure 1). The capacity of microorganisms to generate IAA is a significant feature in promoting plant growth. Endophytic fungi like *Aspergillus*, *Chaetomium*, *Exophiala*, *Fusarium*, *Paecilomyces*, *Penicillium,* and *Phoma* have been demonstrated to produce IAA [41]. IAA can stimulate lateral roots’ formation and root hairs’ development, ultimately enhancing the ability of nutrient absorption in plants [13,42]. Previous work has shown that the solubilization of phosphorus in soil is closely linked to microbial metabolites and low-molecular-weight organic acids released during metabolism [43]. The utilization of phosphate-solubilizing strains has significantly enhanced phosphate uptake, plant growth, and crop yield across various agricultural crops [44]. The solubilization of inorganic phosphate enhances the solubility of soil fixed phosphorus, which is particularly beneficial in conditions where phosphate is a limiting factor. In this study, six endophytic fungi strains were screened with phosphorus soluble capacity ranging from 47.32 to 125.95 μg/mL (Figure 2). The plant growth-promoting strain could enhance iron uptake, leading to increased resilience to abiotic stress [45]. Among the 14 fungal endophytes tested, seven endophytic fungi strains could produce siderophores (Table 2). Chowdappa et al. [46] found that 60% of the endophytic fungi found in *Cymbidium aloifolium* (L.) Sw. produced hydroxamate siderophores and demonstrated high antibacterial activity against key virulent phytopathogens. However, certain strains did not exhibit the ability of siderophore production, potentially because of the lack of specific siderophore biosynthetic genes [47]. In addition, in this study, eight endophytic fungi exhibited nitrogen fixation ability, three fungi showed potassium-dissolving ability, and five fungi were capable of producing ammonia (Figure 3). Nitrogen and potassium are crucial for enhancing plant photosynthesis [1]. Endophytic fungi can provide plants with ample amounts of ammonia to support root and bud elongation [48]. Depending on PGP traits, we found that *Fusarium* sp. ZZ13 and *Alternaria* sp. ZZ10 have multiple PGP abilities, containing IAA production, nitrogen fixation, phosphate solubilization, siderophore production, and ammonia production. Liu et al. [49] showed endophytic microbes from Taxus yunnanensis (Chinese Yew) selected based on producing multiple PGP characteristics could promote the growth of the *Arabidopsis thaliana.* Also, inoculation of rhizospheric and endophytic bacteria with multiple PGP properties showed higher promoting ability [50].

Tobacco is not only an economic crop but is also recognized as a significant model organism in plant biology research [51,52]. CX is difficult in cultivation and is affected by the planting season, so tobacco was chosen to investigate the impact of endophytic fungi on seed germination and seedlings growth. The results showed that only four endophytic fungal strains, including YMY5, YMY6, ZZ10, and ZZ13, were successfully co-cultured with tobacco in MS agar medium and promoted the growth of tobacco. The co-cultivation of these four endophytes with tobacco seedlings enhanced the fresh weight and lateral root number, but the length of primary roots was reduced (Figure 4). The modifications of root structure can be linked to IAA produced by endophytic fungi [1]. Wang et al. [53] reported that endophytic fungus strain TH15 from *Tetrastigma hemsleyanum* produced IAA, methyl-IAA ester (MeIAA), jasmonic acid (JA), jasmonoyl-L-isoleucine (JA-ILE), and salicylic acid (SA), which regulated the root development of *Tetrastigma hemsleyanum.* And beyond that, microorganisms can alter root–system architecture by emitting volatile organic compounds (VOCs) and fungal metabolites [54,55]. 

These four endophytic fungi strain (YMY5, YMY6, ZZ10, and ZZ13), were chosen as inoculants for seed germination and pot experiments in the subsequent experiments due to their ideal PGP characteristics and successful symbiotic relationship with tobacco seedlings. Many endophytic fungi exhibited the capacity to stimulate seed germination and seedling growth [56]. This study demonstrated that ZZ10 and ZZ13 significantly promoted the germination potential, germination index and vigor index of tobacco, while the effect on germination rate was not significant (Figure 5). Agbodjato et al. [57] reported similar findings, demonstrating that the combination of rhizobacteria strains significantly enhanced the seeds vigor index, showing an increase of up to 36.44%. According to El-Nagar et al. [58], the germination of maize seeds was substantially elevated following treatment with *Alternaria alternate*, *Aspergilus flavus*, and *Aspergillus terreus*. Bio-priming can enhance seed vigor even under unfavorable conditions. Fungal endophytes could improve *Achnatherum inebrians* seeds germination under abiotic stress conditions [59]. The enhancement of seed germination and plant growth parameters by endophytic fungi may be linked to their production of plant hormones [60]. 

The growth-promoting properties of endophytic fungi can supply plants with more balanced nutrients. The findings demonstrated that the plant length, fresh weight, and maximum leaf area of tobacco were significantly elevated following the inoculation of ZZ10 and ZZ13. Effects of strain ZZ13 on plant growth was the most significant (Figure 6C–E). Similarly, Zhang et al. [61] found endophytic fungus *Talaromyces muroii* SD1–4 increased growth parameters, including plant height, root length, fresh weight, and dry weight. The colonization of strains was a key step in facilitating plant growth, particularly with regard to the capability of endophytic fungi to colonize non-host plants [62]. In this study, the mycelium wrapped around the surface and inner layers of the tobacco roots in the inoculation group, indicating that endophytic fungi could colonize successfully and effectively promote plant growth (Figure 6B). Shan et al. [63] discovered that the fungus *Paraphaosphaeria* sp. JRF11 effectively colonized and facilitated the growth of tomato. Hu et al. [64] found that the early colonization of roots by *Metarhizium* and *Pochonia* could improve the stem length, stem weight, and root weight of hemp. The colonization is correlated with growing conditions, host interaction, light conditions, temperature, and nutrient competition among soil microbes [65]. The effective colonization of endophytic fungi is necessary for promoting promote plant growth [66].

Chlorophyll is very important pigment for increasing the photosynthetic rate of plants. The content of chlorophyll can elevate the photosynthetic rate of plants. Previous research has indicated that a combination of fungi and rhizobacteria as inoculants can boost photosynthesis efficiency and the levels of photosynthetic pigments [67]. Wang et al. [68] found that the Pn, Gs, and total chlorophyll content in lettuce and celery were observably improved after the treatment with microbial inoculants. In this study, inoculation with YMY5 significantly increased the chlorophyll a and total chlorophyll content. YMY6, ZZ10, and ZZ13 significantly improved the net photosynthetic rates in tobacco leaves, but inoculation of the four endophytic fungi had no significant effect on the rate of stomatal conductance (Gs), transpiration rate (Tr), and intercellular CO_2_ concentration (Ci) (Figure 7). The improvement of photosynthetic performance contributed to laying a crucial groundwork for increasing biomass. The increase of chlorophyll content enhanced the fixation of CO_2_ and improved photosynthetic performance. Previous studies showed that an increase in antioxidant enzyme activity reduces levels of ROS and H_2_O_2_ in guard cells, leading to open the stomata, improved Gs, and enhanced Pn and Tr [69]. 

The content of soluble sugar and soluble protein can be used as an important physiological index to evaluate plants health, adaptability, and growth state. These compounds can decrease cell osmotic pressure and enhance plants resistance to adversity. The study demonstrated that inoculation with YMY5, ZZ10, and ZZ13 elevated levels of soluble sugar and soluble protein in tobacco leaves (Figure 8). The rapid response of plant antioxidant defense system in response to stress stimuli prior to any signs of harm is essential for plants to thrive in harsh environmental settings [69]. Some research has indicated that endophytic fungi could regulate the activity of antioxidant enzymes in plants. According to Li et al. [70], the inoculation of nine endophytic fungi markedly raised the activities of SOD, PAL, CAT, and POD, and proline content, and reduced MDA contents in *Salvia miltiorrhiza Bunge*. Consistent with prior research, this study showed that inoculation with four endophytic fungi elevated CAT and POD activities. In addition, YMY5 remarkably elevated SOD activities, and ZZ10 and ZZ13 lowered MDA content in tobacco leaves (Figure 9). Therefore, endophytic fungi from CX can boost the activity of tobacco antioxidant enzymes and thus enhanced the resilience of plants to adverse environmental conditions, which was critical for growth of plants.

Endophytic fungi can boost the expression of hormone-related and growth-related genes in plants. They upregulated the expression of genes involved in auxin biosynthesis and metabolism [71]. Xu et al. [72] demonstrated that cotton treated with *Bacillus paralicheniformis* RP01 elevated amounts of brassinosteroid (BR), IAA, salicylic acid (SA), and jasmonic acid (JA). Aktar et al. [73] demonstrated that endophytic bacteria could facilitate plant growth by upregulating the expression of gibberellin gene. According to the results of this research, the expression of all measured hormones and growth-related genes increased following ZZ13 inoculation relative to the control group. The expression levels of IAA, BR, GA, and cell cycle genes were increased, and the expression of defense-regulating synthetic genes SA were also increased (Figure 10). Association between plants and endophytes requires interaction between plant hormones [74]. The pivotal role of auxin in facilitating beneficial root-microbial interactions suggests its contribution to maintaining a balance between growth and defense responses [75].

## 5. Conclusions

In this study, multiple potential PGP characteristics of the endophytic fungi of *Ligusticum chuanxiong* from CX and their plant growth promotion effects were analyzed. Each of the 14 strains had one or more PGP properties. The PGP properties of the endophytic fungi of CX indicated that they may improve the nutrient uptake and availability of inorganic elements in soil. In addition, four promising strains belonging to *Fusarium asiaticum* YMY5, *Colletotrichum camelliae* YMY6, *Alternaria alternata* ZZ10, and *Fusarium tricinctum* ZZ13 enhanced various physiological properties in tobacco, including root structure, seed germination, biomass, soluble sugar and protein content, photosynthetic characteristics, and stress resistance. This suggests their potential as effective biofertilizers or plant growth regulators in sustainable agriculture. The study also found that endophytic fungi regulated tobacco growth and defense responses by increasing the expression of plant hormone synthesis genes and cell cycle related genes. In particular, *Fusarium tricinctum* ZZ13 had the greatest effect on the vegetative growth and metabolism of tobacco. This research lays a solid theoretical foundation and offers technical reference for the production of microbial agents from endophytic fungi from CX.

## Figures and Tables

**Figure 1 jof-10-00713-f001:**
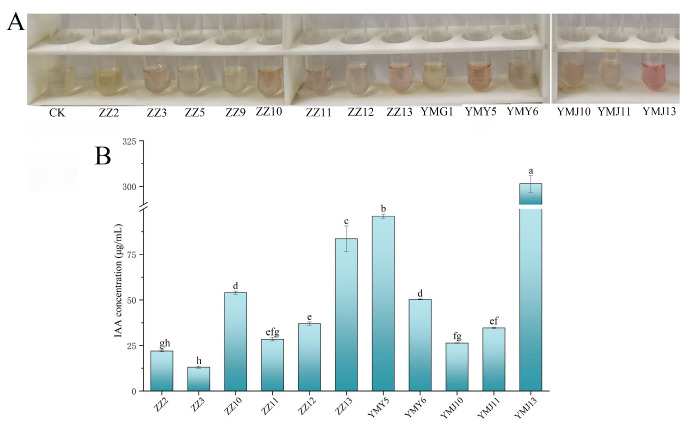
(**A**) Color reaction of 14 endophytic fungal strains for IAA production; (**B**) concentration analysis of IAA production in endophytic fungal strains. Data are represented as mean ± SD of triplicate samples. Varying letters reflect a significant statistical variation at the *p* < 0.05 level.

**Figure 2 jof-10-00713-f002:**
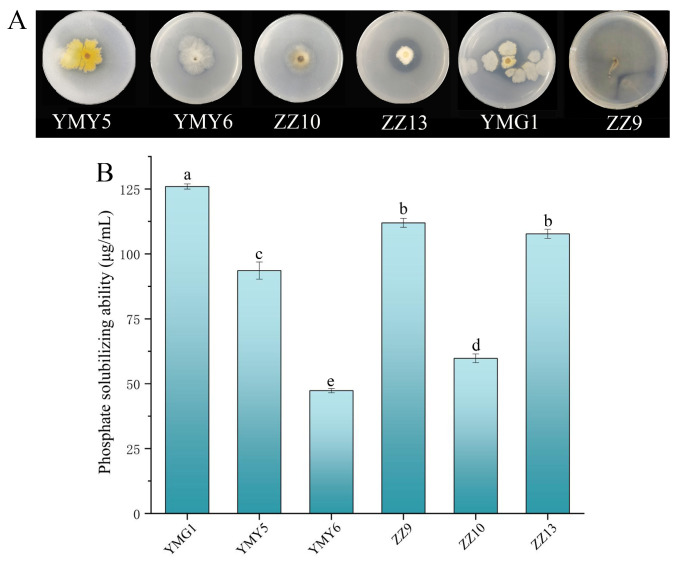
(**A**) Six endophytic fungal strains exhibited transparent halos around their colonies on the NBRIP medium. (**B**) The capacity of inorganic phosphate-solubilizing of the six strains. Data represented as mean ± SD of triplicate samples. Varying letters reflect a significant statistical variation at the *p* < 0.05 level.

**Figure 3 jof-10-00713-f003:**
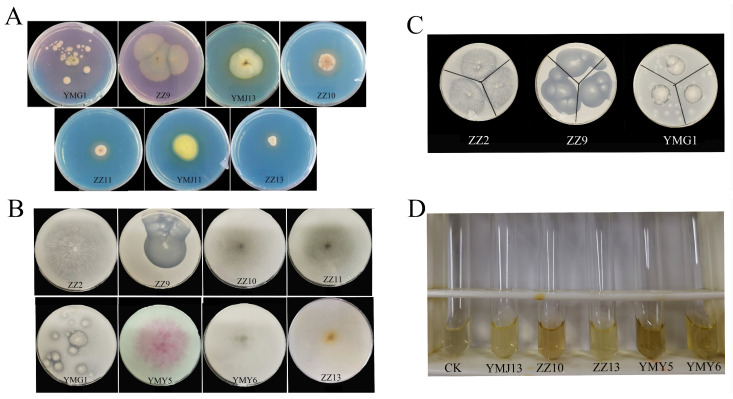
The positive results of endophytic fungal strains to produce siderophore (**A**), nitrogen fixation (**B**), potassium solubilization (**C**), and ammonia production (**D**).

**Figure 4 jof-10-00713-f004:**
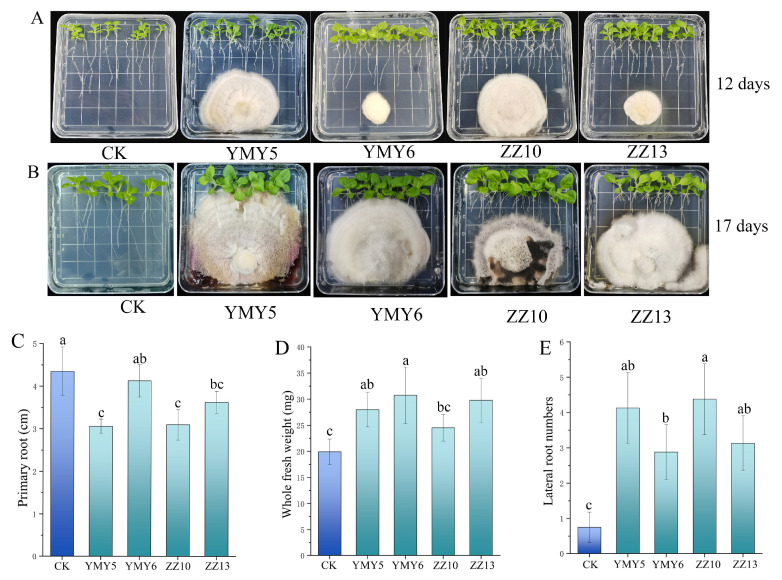
Representative photographs of tobacco co-cultured with endophytic fungi of CX for (**A**) 12 days (**B**) and 17 are were shown. (**C**) Primary root length, (**D**) total fresh weight, and (**E**) lateral root number of tobacco were measured on the 12th day after treatment with endophytic fungi. Data are represented as mean ± SD. *n* = 8. Varying letters reflect a significant statistical variation at the *p* < 0.05 level.

**Figure 5 jof-10-00713-f005:**
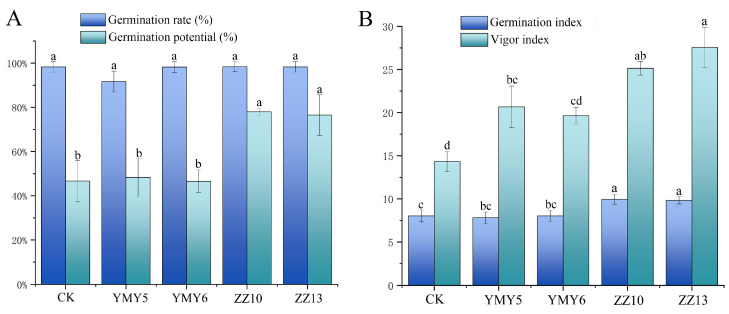
Effects of four endophytic fungi on (**A**) germination rate and germination potential and (**B**) germination index and vigor index of tobacco seed. Data represented as mean ± SD. *n* = 20. Varying letters reflect a significant statistical variation at the *p* < 0.05 level.

**Figure 6 jof-10-00713-f006:**
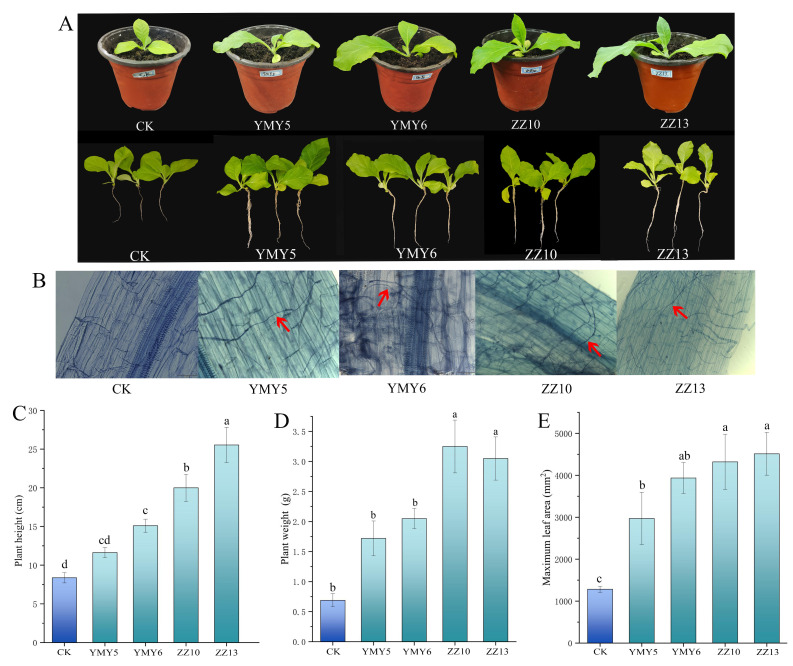
The impact of inoculating tobacco with four endophytic fungi on the growth dynamics of tobacco. (**A**) Morphology of tobacco seedlings after 50 days of treatment; (**B**) the colonization of endophytic fungi in tobacco roots; (**C**) plant height; (**D**) plant fresh weight; and (**E**) maximum leaf area. Data are represented as mean ± SD of triplicate sample. Varying letters reflect a significant statistical variation at the *p* < 0.05. The hypha of fungal endophytes colonizing in the root of tobacco are represented by red arrows.

**Figure 7 jof-10-00713-f007:**
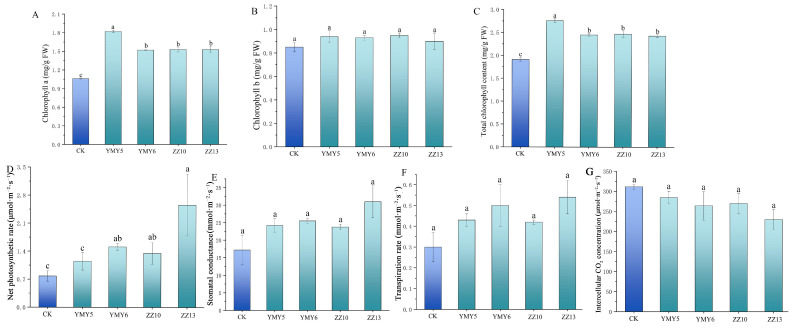
The impact of endophytic fungi on chlorophyll content and photosynthesis parameters of tobacco in pot experiment. (**A**) chlorophyll a; (**B**) chlorophyll b; (**C**) total chlorophyll; (**D**) net photosynthetic rate (Pn); (**E**) stomatal conductance (Gs); (**F**) transpiration rate (Tr); and (**G**) intercellular CO_2_ concentration (Ci). Data reflect the mean values calculated from three separate replicates. Varying letters reflect a statistically significant variation at *p* < 0.05.

**Figure 8 jof-10-00713-f008:**
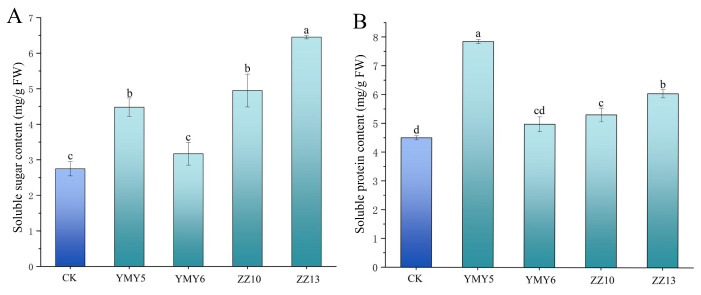
The inoculation of tobacco with four endophytic fungi influences (**A**) soluble sugar and (**B**) soluble protein content in the leaves. Data reflect the mean values calculated from three separate replicates. Varying letters reflect a statistically significant variation at *p* < 0.05 level.

**Figure 9 jof-10-00713-f009:**
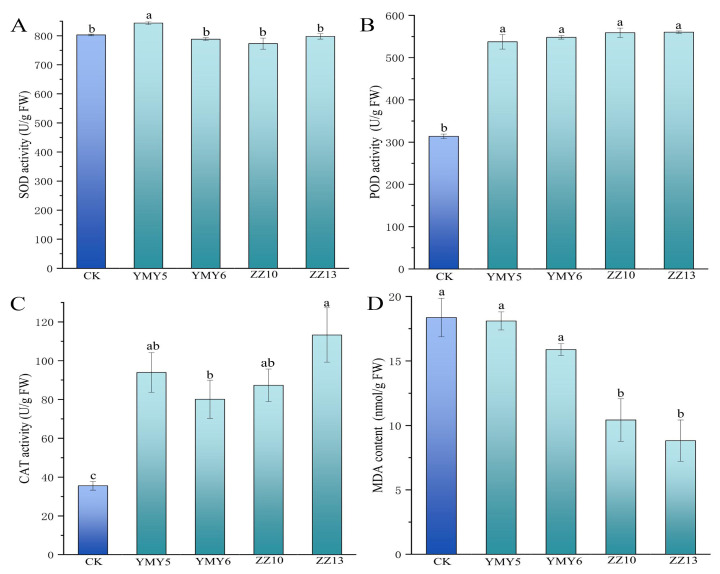
The influence of four endophytic fungi on the antioxidant enzyme activity of tobacco leaves. (**A**) SOD activity; (**B**) POD activity; (**C**) CAT activity; and (**D**) MDA content. Data reflect the mean values calculated from three separate replicates. Varying letters reflect a statistically significant variation at *p* < 0.05 level.

**Figure 10 jof-10-00713-f010:**
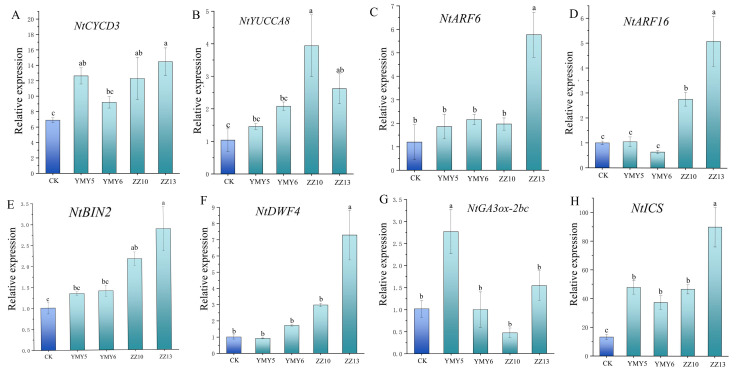
Effects of endophytic fungi on phytohormone-related gene expression in tobacco leaves. (**A**) *NtCYCD3* (cell cycle-related); (**B**) *NtYUCCA8* (IAA biosynthesis gene); (**C**) *NtARF6* and (**D**) *NtARF16* (auxin response factors); (**E**) *NtBIN2* (the BR signaling genes); (**F**) *NtDWF4* (the BR biosynthesis-related genes); (**G**) *NtGA3-ox2* (GA_3_ biosynthesis gene); (**H**) *NtICS* (SA biosynthesis gene). Data reflect the mean values calculated from three separate replicates. Varying letters reflect a statistically significant variation at *p* < 0.05.

**Table 1 jof-10-00713-t001:** The genetic primer sequence associated with tobacco growth.

Gene	Gene Accession	Sequence	Primers	Reference Gene
*NtCYCD3*	XM_016602338.1	F′	AGAGAGGCCGTTGATTGGAT	Cell cycle-related
		R′	GAAAGACAGGACACAGCAGC	
*NtARF6*	XM_016594427.1	F′	CCCACTACTTATTTGCCAGCTT	Auxin response factors
		R′	AGATGCCTCCTTTTGCTCT	
*NtARF16*	XM_016609598.1	F′	ACCTGAGCTAAGTACCGTAGAA	Auxin response factors
		R′	GCTTGCGGTGAAGAAATTGAG	
*NtGA3ox-2*	EF471116	F′	TGCTCGGCCCTACAACTAAA	GA_3_ biosynthesis gene
		R′	AACCGTGACCCAACCATTTC	
*NtDWF4*	XM_016618664.1	F′	CTGCCAGCCTTTGGTAATTTG	BR biosynthesis-related genes
		R′	CACTTCAAACCGTTCGTCATTT	
*NtBIN2*	XM_016592263.1	F′	CTCATACATCTGCTCCCGGT	BR signalling genes
		R′	ATTTCCTCCCTTGTCGGTGT	
*NtYUCCA8*	XM_016592388	F′	ATGTGTATGGGTAAATGGTCC	IAA biosynthesis gene
		R′	CAGATTTTTCCAAGATTACAC	
*NtICS*	XM_016616491.1	F′	ATGTATGCTGGTCCTGTTG	SA biosynthesis gene
		R′	AATCACTTCCTTCCACTATCC	
*β-Actin*	AB158612	F′	GATCTTGCTGGTCGTGATCT	Reference gene
		R′	ACTTCCGGACATCTGAACCT	

**Table 2 jof-10-00713-t002:** Qualitative analysis of PGP activities including siderophores, nitrogen fixation, ammonia production, and potassium-solubilizing activities. “+” shows a positive reaction, while “−” represents a negative reaction.

Strains	Siderophores	Nitrogen Fixation	Ammonia Production	Potassium Solubilization
ZZ2	−	+	−	+
ZZ3	−	−	−	−
ZZ5	−	−	−	−
ZZ9	+	+	−	+
ZZ10	+	+	+	−
ZZ11	+	+	−	−
ZZ12	−	−	−	−
ZZ13	+	+	+	−
YMG1	+	+	−	+
YMY5	+	+	+	−
YMY6	+	+	+	−
YMJ10	−	−	−	−
YMJ11	+	−	−	−
YMJ13	+	−	+	−

## Data Availability

Data are contained within the article.

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
