# Peer review of "Exploring Plant Growth-Promoting Traits of Endophytic Fungi Isolated from Ligusticum chuanxiong Hort and Their Interaction in Plant Growth and Development"

_jof, 2024, doi:10.3390/jof10100713_

Round 1
Reviewer 1 Report
The manuscript is devoted to the actual topic of searching for endophytic fungi with growth-stimulating properties and studying their effect on the growth and development of agricultural crops. . This can provide a basis for creating new biofertilizers and biostimulants and ensure the development of sustainable crop production.
1. The Abstract should indicate how many endophyte isolates were studied.
2. The keywords are too general; they should reflect what the authors specifically did.
3. Lines 65-68. A reference to the source of information should be added.
4. Lines 68-70. A reference to the source of information should be added.
5. Expand the abbreviations PDA, BR, GA.
6. Specify the compositions of all media for culturing fungi.
7. Lines 121-122. Repeat the text on lines 119-120.
8. I recommend that the authors include a table in the Results section indicating the functions of the genes whose expression was studied by the authors.
9. I recommend that the Discussion section include links to the figures and the table.
It is questionable whether the differences between treatment groups in Fig. 1B, 3(C, D, E), 7(D, E, F, G), 8B, 9(A, D), 10(A, B, D, E, F) are statistically significant. The correctness of the arrangement of the letters above the columns in these diagrams is also questionable. The authors should check the statistical processing of the data.
11. The Conclusion should not describe the results obtained once again. It should be more general in nature and indicate the direction in which the authors' further research will develop.
Reviewer 2 Report
Dear Authors!
The manuscript is clear, relevant for the field and presented in a well-structured manner. The cited references are mostly recent publications and relevant. It does not include an excessive number of self-citations. The manuscript is scientifically sound and the experimental design is appropriate to test the hypothesis. The manuscript’s results are reproducible based on the details given in the methods section, but you should extend it. Write more about germination characteristics, about the composition of the mediums.
The figures, tables, images are appropriate. They properly show the data. They are easy to interpret and understand. Check statistical analysis in Figure 5E.
The conclusions are consistent with the evidence and arguments presented. I didn`t noticed any ethics conflicts in the text of the manuscript.
1) Correct 1-amino-1-carboxylic acid to 1-aminocyclopropane-1-carboxylic acid (line 47).
2) Change the words producers, producing to production (lines 87-88).
3) Decipher for the first time NBRIP (line 118), CAS (line 131), MeIAA, JA, JA-ILE (line 508).
4) The sentences “Endophytic fungi were grown on NBRIP agar medium for 7 d at 28°C to observe the formation of a transparent zones surrounding colonies” (lines 119-120) and “Fungal strains were inoculated into NBRIP liquid medium at 28°C for 7 d” (lines 121-122) have the same meaning. Rephrase one of them.
5) Check the formula FeCl (line 134).
6) Write more about the substances in the Aleksandrov agar medium (line 142), Murashige and Skoog medium (156). Write the origin of Nessler's reagent (line 150).
7) Check the name of centrifugation parameter r/min (lines 200, 220).
8) Add the concentrations of sulfuric acid and anthrone (lines 215-216) in the text.
9) Change the words “Quantification of Resistant…” to “Quantification of Activity of Antioxidant Enzymes” (line 218), mol/L to M (line 219). MDA is not an enzyme (line 222), so you should write about its estimation separately.
10) Clarify what rate is being described on line 361.
11) Please, clarify if soluble sugar and soluble protein are really secondary metabolites (line 407).
12) Did you calculate soluble sugar and soluble protein contents per g FW? Add information about this in Figure 8.
13) Remove information about cell membrane damage (MDA) from lines 432-435 to line 418 where it was first mentioned.
14) Change the word protective to antioxidant (line 567).
Reviewer 3 Report
This article explores the PGP capabilities of 14 endophytic fungi previously isolated from roots of the cultivated species Ligusticum chuanxiong, as well as the influence of 4 of these fungi on tobacco plant growth, seed germination, photosynthetic traits, sugar content and expression of growth-related genes. The differences between inoculated and non-inoculated plants highlight the potential relevance of these fungi as PGP microorganisms. This issue is relevant and timely for L. chuanxiong crops but also for other crops.
However, the interest of the study is limited by the poor organization of the Introduction and Discussion sections, with a poor review of previous works. Both sections are mainly built with constant repetitions of concepts and both are full of generalistic or superficial statement that do not go into depth in the exploration/discussion of the results or the previous research.
Besides, in the Introduction section no research gap is clearly presented. No research hypotheses are stated.
In the Discussion section, the studied endophytes are barely discussed in the perspective of their known traits, and comparative data of these species/genera from other studies are barely presented.
The applicability and generalizability of the results is seriously called into question by the defficiencies in the statistical analysis of the data. The sample size of the study is very small (3 repetitions for each treatment in some cases and 1 sample per treatment in a few others). Most of the study can only be considered as exploratory results, but this is not acknowledged anywhere.
In some experiments (Fig. 1, 2, 6), only one sample per treatment is available. The same sample is measured by triplicate. This is not experimental replication. This is only one sample. No statistic treatment should be used to analyze these results, and data presentation should clearly acknowledge that the discussion is based on only one sample per treatment.
In the remaining experiments, the results are presented as mean + SD. The standard deviation is not a proper way of presenting variability in a manipulative experiment. The confidence interval is.
In the Material section, no mention to the statistical procedure used for analyzing the experiments is done (do you use ANOVA?). What about the checking of the ANOVA assumptions? No data transformations were necessary?
In the results section, wherever an statistical analysis is performed, the F-value, p-value and n should be presented for each ANOVA.
The experimental design and/or sample size is not explained in sections 2.2-2.10. How many fungi are studied? How many samples per fungi? Only one treatment per fungi? Control treatments in 2.3?
The manuscript is also hindered by several minor issues:
- L70-71 It is said that "In conclusion, the efficient utilization of endophytic fungi holds significant practical implications for crop yield and quality". However, in the Introduction no review of other crops is presented.
- L75 "expelling wind". If the species is used in traditional Chinese medicine, it should be clearly stated.
- L75-76 It is said that "Endophytic fungi and their secondary metabolites of CX have already been thoroughly investigated". Then, why is this study necessary? Where are the references of these abundant papers? Only three are cited in the Introduction.
- L82-84. Reference 17 does not allow to doubtlessly conclude that.
- L84 It is said "Despite the endophytic fungi from CX are promising". This sentence does not make sense.
- L181. Which kind of pots are used? Which volume? What does "grown under natural conditions" mean? Which is the mycelium application dose? Which are the characteristics of the used soil? Which sterilization method was used?
- L168 How was the spore suspension produced?
- The experimental design and/or sample size is not explained in experiments 2.2-2.10. How many fungi are studied? How many samples per fungi? Only one treatment per fungi? Control treatments in 2.3?
- The Results section include many sentences that would more properly be located in either the Introduction or the Discussion. For example, L249-251 (first sentence in the section), L256, L271-275, L298, L303, L370-373 and a few more.
- L253 qualitative analysis?
- Fig 2 Why are not the rest of the species included in the figure? And the control values?
- Fig 5 Which is the sample size? Which are the units in subfig 5B?
- L370 How many tips per seedling were observed?
- Please, include in the Materials section how the germination potential, germination index and vigor index were calculated.
- Fig 6D. Fresh or dry weight?
- L389. It is said "To conclude, the findings demonstrated that inoculating with endophytic fungi enhanced the content of chlorophyll a, chlorophyll b and total chlorophyll of tobacco". However, according to your statistic analysis, chlorophyll b did not significantly change.
- Fig 7. Please include sample size and avoid abbreviations in figure axes
- L472 plant growth and quality. What does "plant quality" mean?
- The study should recognize its limitations, such as its exploratory character for most of the studied variables (due to the lack of experimental replication or low sample size), the fact that effects should be tested in field conditions to confirm the potential use of these fungi as PGP in agriculture, and the fact that effects are only tested in the very first stage of growth (50 days) of one plant species.
Reviewer 4 Report
Dear Editor
Many thanks for considering me as a potential reviewer for the article "Multiple Potential Plant Growth Promotion Traits of Endophytic Fungi Isolated from Ligusticum chuanxiong Hort and Their Interaction in Plant Growth and Development". No doubt the article is well-structured, well-presented and written. However, I have several observations that should be considered before proceeding further.
1, English is very difficult to understand, needs extensive language editing,
2, The article introduction is poorly cited,
My detailed comments can be found below,
Thanks
Dear Editor
Many thanks for considering me as a potential reviewer for the article "Multiple Potential Plant Growth Promotion Traits of Endophytic Fungi Isolated from Ligusticum chuanxiong Hort and Their Interaction in Plant Growth and Development". No doubt the article is well-structured, well-presented and written. However, I have several observations that should be considered before proceeding further.
My observations are as follows.
My observations
Title
· I will suggest shortening the title, please. For example, ‘Multiple Potential’ seems repetitive, just a suggestion (you can write another one) is ‘Exploring/Investigating Plant Growth-Promoting Traits of Endophytic Fungi Isolated from Ligusticum chuanxiong Hort and Their Interaction in Plant Development’
· Another thing, ‘Ligusticum chuanxiong’ should be italicized (do the said throughout the manuscript).
Introduction
· Line-33 ‘with their hosts[1]’ there must be a space like, ‘with their hosts [1]’, please do the said throughout the manuscript,
· Lines 39-42, Where is numerous citations? Numerous studies have consistently indicated the pivotal role of endophytic fungi in ecosystem dynamics, elucidating their crucial functions in facilitating the growth of plants. Therefore, endophytic fungi have great research and development potential across various domains, such as agriculture and forestry.
· Line-46, please pay attention to the minor issues like phytohormone should be phytohormones,
· Line-44-47 lack citations ‘Through a combination of direct and indirect pathways, endophytic fungi facilitate plant growth. The direct mechanism involves the stimulation or induction of plants to synthesize specific bioactive substances, like the production of phytohormone, siderophore, ACC (1-amino-1-carboxylic acid) deaminase, nitrogen fixation, phosphorus solubilization and potassium solubilization,
· Please give one/two examples to support your idea/work, for example, isolation of endophytes from closely related host plants, selection of effective strains and different traits improvements/reduction after inoculation to seedling/seeds.
· In general, the introduction I poorly cited, please cite each information, accordingly.
Material and method
· What do you mean by ‘2.1. Materials’?, please select another phrase like Endophytic strains selection etc…..
· Line-110 please re-write the sentence again ‘After 110 centrifuging the culture for 30 min at 6000 rpm’,
· In section statistical analysis; A) Which test was applied? B) ‘P<0.05’ p should be italicized (do the said throughout the manuscript),
Results
· Line-271-274 This information should go to discussion ‘Previous work shown that the solubilization of phosphorus in soil was closely linked to microbial metabolites and low molecular weight organic acids released during metabolism[31]. The utilization of phosphate-solubilizing strains significantly enhanced phosphate uptake, plant growth, and crop yield across various agricultural crops[32].,
· Please don’t need to justify your findings in results it will be done in the discussion, please remove such justifications and put them in the discussion,
· Figure-4E (A) The statistical letters on the graph are wrong please pay attention, (B) Please increase the sizes of each statistical letter, very difficult to read,
Discussion
Please pay attention to your results and compare them with previous findings.
Round 2
Reviewer 1 Report
I still believe that the differences between groups in Figs. 1B, 3(C, D, E), 7(D, E, F, G), 8B, 9(A, D), 10(A, B, D, E, F) are statistically insignificant. The correctness of the placement of the letters above the bars in these diagrams is also questionable. The Statistical Analysis section does not indicate how the data are presented. Is this mean ± standard error?
Dear Authors!
I have received answers to my questions and comments #1-9 and #11. However, despite your comments that you checked the statistical processing of the data, this was not done. I still believe that the differences between groups in Figs. 1B, 3(C, D, E), 7(D, E, F, G), 8B, 9(A, D), 10(A, B, D, E, F) are statistically insignificant. The correctness of the placement of the letters above the bars in these diagrams is also questionable. The Statistical Analysis section does not indicate how the data are presented. Is this mean ± standard error?
Author Response
Comment: I still believe that the differences between groups in Figs. 1B, 3(C, D, E), 7(D, E, F, G), 8B, 9(A, D), 10(A, B, D, E, F) are statistically insignificant. The correctness of the placement of the letters above the bars in these diagrams is also questionable. The Statistical Analysis section does not indicate how the data are presented. Is this mean ± standard error?
Recover: We deeply appreciate the reviewer’s suggestion. We re-analyzed the data by one-way analysis of variance (ANOVA) followed by Tukey's test using the SPSS24.0 software. Based on the existing statistical data analysis, we reinterpreted the significant results in the article. “All results were presented in terms of mean ± standard deviation (SD)” was added to the Statistical Analyses.
Reviewer 2 Report
Dear Authors!
This is my second review to this manuscript, so I will not write a lot of words. You changed the text according my recommendations, the article became better.
My comments:
Keywords will help to search your article, so I think you should write them more precisely. Change "physiological characteristics" to "phytohormones-related genes, antioxidant enzymes, photosynthesis parameters, soluble sugars and proteins".
I think, you should decipher the abbreviations in Abstract.
Author Response
Comment: Keywords will help to search your article, so I think you should write them more precisely. Change "physiological characteristics" to "phytohormones-related genes, antioxidant enzymes, photosynthesis parameters, soluble sugars and proteins". I think, you should decipher the abbreviations in Abstract.
Recover: Thank you for finding this detail. We deeply appreciate the reviewer’s suggestion and agree with the comment. Due to the limitation of 3-5 keywords, it is impossible to include all physiological indicators at the same time. We believe that we can choose some core keywords to represent these. We deciphered the abbreviations in Abstract.
Reviewer 3 Report
The minor comments in the revision were adressed. However, several major issues detailed in the revision were not:
1) The results are presented as mean + SD. The standard deviation is not a proper way of presenting variability in a manipulative experiment. The confidence interval is.
2) No information about the checking of the ANOVA assumptions is provided.
3) In the text, the F-value and n should be presented for each ANOVA.
4) In the introduction section, no research hypotheses are stated
The minor comments in the revision were adressed.
Author Response
Comment 1: The results are presented as mean + SD. The standard deviation is not a proper way of presenting variability in a manipulative experiment. The confidence interval is.
Recover 1: Thank you for your comments. We describe the analysis method in detail in the data analysis.
Comment 2: No information about the checking of the ANOVA assumptions is provided.
Recover 2: Thank you for finding this detail. We describe the analysis method in detail in the data analysis. The results were subjected to an analysis of variance (ANOVA) when the assumptions of normality and variance homogeneity were met. We re-analyzed the data by one-way analysis of variance (ANOVA) followed by Tukey's test using the SPSS24.0 software.
Comment 3: In the text, the F-value and n should be presented for each ANOVA.
Recover 3: Thank you for finding this detail. we did not find such a presentation of the F-value in other literature; therefore, I'm sorry we didn't revise this opinion. The sample size (n) can be found in the figure caption and the methods section.
Comment 4: In the introduction section, no research hypotheses are stated
Recover 4: Thank you for finding this detail. We added to the possible effects of inoculation with endophytic fungi on tobacco growth and physiology in the introduction section.
Reviewer 4 Report
Dear Editors/Authors,
Many thanks for your consideration of the point-to-point response and corrections!
The article is fine, however, please add authority names to all scientific names, especially plants names,
If possible, please improve the quality of your figures.
Several issues/errors are detected, regarding the English language!
Thanks
Dear Editors/Authors,
Many thanks for your consideration of the point-to-point response and corrections!
Here are a few corrections, please have a look,
Line-106: Nicotiana tabacum lacking authority name, please check whole MS and do the said for all plants names (Taxus yunnanensis........(line 527)
Please improve the figures/graphs' qualities,
Figure 6B, What do the red arrows indicate?, please indicate,
Lines 417-418: no need to justify your findings here, this information should go to discussion (The change of Pn, Tr, and Gs in tobacco following inoculation correspond with the findings 418 from earlier report [41].
Thanks
Author Response
Comment 1: Line-106: Nicotiana tabacum lacking authority name, please check whole MS and do the said for all plants names (Taxus yunnanensis........(line 527)
Recover 1: Thank you for your feedback regarding the manuscript. I have made the necessary modifications and have checked the entire manuscript.
Comment 2: Please improve the figures/graphs' qualities, Figure 6B, What do the red arrows indicate? please indicate,
Recover 2: Thank you for finding this detail. We have added "The hypha of fungal endophytes colonizing in the root of tobacco were represented by red arrows" in Figure 6.
Comment 3: Lines 417-418: no need to justify your findings here, this information should go to discussion (The change of Pn, Tr, and Gs in tobacco following inoculation correspond with the findings 418 from earlier report [41].
Recover 3: Thank you for your comments. We deeply appreciate the reviewer’s suggestion. We have deleted this sentence in lines 417-418 as per your suggestion and moved the justification of the findings to the discussion section.
Round 3
Reviewer 1 Report
Dear Authors!
Thank you, now the statistical processing has been carried out qualitatively. The differences between the compared groups are reliable.
Dear Authors!
Thank you, now the statistical processing has been carried out qualitatively. The differences between the compared groups are reliable.